# Mantle cell lymphoma with gastrointestinal involvement and the role of endoscopic examinations

Han Hee Lee[1], Seok-Goo Cho[2]*, In Seok Lee[1]*, Hye Jin Cho[1], Young-Woo Jeon[2], Joo Hyun O[3], Seung Eun Jung[4], Byung Ock Choi[5], Kyung-Sin Park[6], Suk-Woo Yang[7]

1 Department of Gastroenterology, Seoul St. Mary's Hospital, College of Medicine, The Catholic University of Korea, Seoul, Republic of Korea, 2 Department of Hematology, Seoul St. Mary's Hospital, College of Medicine, The Catholic University of Korea, Seoul, Republic of Korea, 3 Department of Nuclear Medicine, Seoul St. Mary's Hospital, College of Medicine, The Catholic University of Korea, Seoul, Republic of Korea, 4 Department of Radiology, Seoul St. Mary's Hospital, College of Medicine, The Catholic University of Korea, Seoul, Republic of Korea, 5 Department of Radiation Oncology, Seoul St. Mary's Hospital, College of Medicine, The Catholic University of Korea, Seoul, Republic of Korea, 6 Department of Hospital Pathology, Seoul St. Mary's Hospital, College of Medicine, The Catholic University of Korea, Seoul, Republic of Korea, 7 Department of Ophthalmology, Seoul St. Mary's Hospital, College of Medicine, The Catholic University of Korea, Seoul, Republic of Korea

* chosg@catholic.ac.kr (SGC); isle@catholic.ac.kr (ISL)

**Data Availability Statement:** All relevant data are within the manuscript and its Supporting Information files.

## Abstract

### Background

Studies on gastrointestinal (GI) tract involvement in mantle cell lymphoma (MCL) are lacking. We investigated the clinical characteristics and prognosis of MCL with GI tract involvement.

### Methods

We retrospectively analyzed 64 patients diagnosed with MCL from January 2009 to April 2017. At the time of MCL diagnosis, patients who were identified to have GI involvement by endoscopic or radiologic examination were assigned to the GI-MCL group. The other patients were assigned to the non GI-MCL group.

### Results

The GI-MCL group included 28 patients (43.8%). The most common endoscopic finding of MCL was lymphomatous polyposis (20/28, 71.4%). The GI-MCL group had higher stage and International Prognostic Index status ($P = 0.012$ and $P = 0.003$, respectively). Among the total 51 GI lesions in the GI-MCL group, 31.4% (16/51) were detected only by endoscopic examinations and were not detected on CT or PET-CT. The cumulative incidence of recurrence was higher in the GI-MCL group compared with the non GI-MCL group but the difference was not statistically significant ($P = 0.082$). Stage (HR 1.994, 95% CI 1.007–3.948) and auto PBSCT (HR 0.133, 95% CI 0.041–0.437) were identified as independent predictive factors for recurrence. Recurrences at GI tract were identified in 59.1% (13/22)

**Funding:** This research was supported by the Bio & Medical Technology Development Program of the National Research Foundation (NRF) funded by the Ministry of Science & ICT (NRF-2018M3A9E8021507). HHL received this grant. The funders had no role in study design, data collection and analysis, decision to publish, or preparation of the manuscript.

**Competing interests:** The authors have declared that no competing interests exist.

and 11.1% (2/18) of the GI-MCL and non GI-MCL group, respectively. Among 15 GI tract recurrences, five recurrences were detected only with endoscopic examinations.

## Conclusions

Endoscopy can reveal the GI involvement of MCL that is not visualized by radiological imaging. Endoscopic examinations are recommended during staging workup and the follow-up period of MCL patients.

## Introduction

Mantle cell lymphoma (MCL) is a type of B-cell non-Hodgkin's lymphoma that is characterized by atypical small lymphoid cells within the mantle zone that surrounds normal germinal center follicles. Although MCL is classified as low-grade lymphoma due to its indolent histological features, MCL is so aggressive that patients usually have advanced stage with massive lymphadenopathy, splenomegaly, and blood and bone marrow involvement at the time of diagnosis [1].

The gastrointestinal (GI) tract is one of common extranodal sites of MCL. While the colon is the most involved site, both the upper and lower GI tract from the stomach to the colon can be involved [2]. Lymphomatous polyposis is the most frequent endoscopic presentation of MCL [3]. However, other shapes including polyp, mass, or even normal appearing mucosa are also presented [3–6].

Although GI tract involvement of MCL was frequently reported up to over 80% microscopically [3, 7, 8], whether GI tract involvement affects the prognosis of MCL patients remains unknown. One study reported the endoscopic and clinical characteristics of 19 MCL patients with visible GI tract involvement. However, there was no comparison with the MCL patients who did not have GI tract involvement and thus determining the clinical significance according to GI tract involvement was not possible [9]. Another study stated that knowledge of GI tract involvement had little impact on patient management decisions despite the high frequency of GI tract involvement [3]. For this reason, the NCCN guideline does not recommend endoscopy or colonoscopy as part of routine initial workup of MCL [10]. However, this recommendation has been hampered by the paucity of studies concerning the oncological outcomes including the response rate of chemotherapy, duration of remission, progression-free survival, and overall survival in MCL patients with GI tract involvement.

This study compared the clinical characteristics of MCL according to the macroscopic GI tract involvement at the time of diagnosis. We also investigated the oncological outcome of MCL and the role of endoscopy to diagnose the disease and detect recurrence in the follow-up of MCL patients.

## Methods

### Patients and eligibility criteria

We retrospectively analyzed patients newly diagnosed with MCL between January 2009 and April 2017 at Seoul St. Mary's Hospital, Seoul, Korea. Pathologic diagnosis was performed according to the current World Health Organization classification by an expert pathologist (Kyung-Sin Park). When the patients were confirmed as MCL pathologically, they underwent the current standard staging workup that included a physical examination, blood cell counts,

routine blood chemistries, computed tomography (CT) of the chest, abdomen, and pelvis, [$^{18}$F]fluorodeoxyglucose-positron emission tomography CT (FDG-PET CT) and a bone marrow evaluation [11]. Among pathologically confirmed MCL patients, those who were checked by radiologic or endoscopic imaging on GI tract were included to determine whether the GI tract was involved. We excluded patients without GI tract images and who were followed up less than 3 months. Patient data were collected from an electronic patient database for the following variables: demographic characteristics, initial diagnosed site, stage, international prognostic index (IPI), MCL international prognostic index with biologic component (MIPIb), treatment, and prognosis. In the MCL group with GI involvement, specific variables for GI lymphoma were also investigated such as presence of GI symptoms, reason of endoscopic examinations, endoscopic finding, and involved site of GI tract. Evaluation and management of MCL patients were performed at the Catholic University Lymphoma Group of Seoul St. Mary's Hospital, which was composed of an expert medical hematologist, endoscopist, radiologist, and pathologist. The Institutional Review Board of Seoul St. Mary's Hospital approved this study (KC11RISI0983). Because of the retrospective and anonymous nature of the data, informed consent was waived.

## Definition

We grouped the patients depending on whether or not the GI tract was affected by MCL. MCL patients with GI involvement were defined as those who had GI involvement of lymphoma that was proven endoscopically and confirmed by biopsy (the GI-MCL group). MCL patients without GI involvement were defined as those whose radiologic or endoscopic imaging results showed no evidence of GI involvement of lymphoma (the non GI-MCL group). The initial diagnosed site indicates the body part at which MCL was initially confirmed by biopsy.

## Treatment and follow-up

All included patients with MCL received induction immunochemotherapy. The regimen of induction immunochemotherapy was determined by discussion of the Catholic University Lymphoma Group. The first line regimens for elderly patients included rituximab plus cyclophosphamide/doxorubicin/oncovin/prednisone (R-CHOP) and bendamustine/ rituximab (BR). For younger patients, rituximab plus hyperfractioned cyclophosphamide/vincristine/ adriamycin/dexamethasone (R-HyperCVAD) or other modified regimens were considered. If remission was not achieved, salvage treatment was considered.

Treatment response was assessed after induction therapy. It consisted of a complete physical examination, determination of blood count and serum lactate dehydrogenase (LDH) level, imaging studies with CT or FDG-PET CT, and endoscopy if indicated. In patients otherwise fulfilling the criteria of complete remission (CR) and with baseline bone marrow involvement, bone marrow aspiration and biopsy was performed. After achieving remission, follow-up assessments included a CT scan were performed every three or six months during the first year and at greater intervals (6–12 months) later. Response was assessed according to the International Working Group criteria for non-Hodgkin's lymphomas [12]. The prognosis of the included patients in this study was followed up until April 2018.

## Statistical analysis

Descriptive statistics were used to characterize the demographic features of the study population. Continuous variables are expressed as means (± standard deviation) or medians (range) and were compared using Student t-tests or Mann-Whitney $U$ test as appropriate. Categorical variables were compared between the groups using the chi-square test or Fisher's exact test as

appropriate. Cumulative incidence of recurrence was calculated from the date of achieving CR until the date of clinical relapse or progression, death from lymphoma, or date of last clinical follow-up. Overall survival was determined as the time between diagnosis and death as a result of any cause and was censored at the latest follow-up of patients who were alive. Survival was estimated using Kaplan-Meier curve analysis, with statistical comparison using the log-rank statistic. The recurrence data were analyzed using univariate and multivariate Cox proportional hazard regression models with a backward elimination procedure to determine the variables that affected recurrence. Variables showing $P$ values <0.20 after univariate analysis and those that were considered clinically relevant were included in a multivariate logistic regression model. Statistical analyses were performed with SAS software package (ver. 8.02, SAS Institute, Cary, NC, USA). In all analyses, a $P$ value <0.05 was considered statistically significant.

## Results

### Patient characteristics

From January 2009 to April 2017, 74 consecutive patients who were diagnosed as MCL were identified (Fig 1). After excluding those who had no GI tract images (n = 7) and those were followed up for less than 3 months (n = 3), 64 patients were included. The numbers of patients in the GI-MCL and non GI-MCL groups were 28 and 36, respectively.

The demographic characteristics of the patient groups are summarized in Table 1. The median age of the GI-MCL group was higher than that of the non-GI MCL group (65 vs. 57 years, $P = 0.044$). The GI-MCL group tended to have higher Ann Arbor stage and IPI status ($P = 0.012$ and $P = 0.003$, respectively). The percentage of high MIPIb risk status was also higher in the GI-MCL group ($P = 0.040$). There were no significant differences in sex, ECOG at diagnosis, and bone marrow involvement between the GI-MCL and non GI-MCL groups.

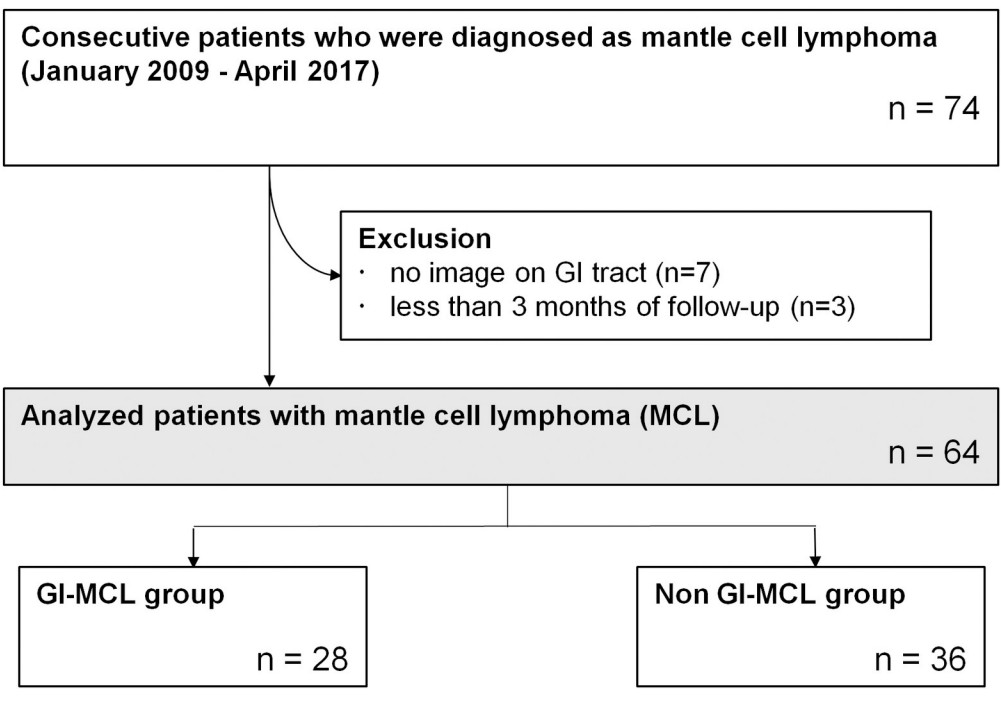

**Fig 1. Flowchart of the study population.**

**Table 1. Demographics and baseline characteristics of included patients according to gastrointestinal tract involvement.**

| N = 64 | GI-MCL | Non GI-MCL | p |
|---|---|---|---|
| | (n = 28) | (n = 36) | |
| Median age, years (range) | 65 (26–83) | 57 (40–77) | 0.044 |
| Male, n (%) | 24 (85.7%) | 29 (80.6%) | 0.835 |
| ECOG at diagnosis | | | 0.127 |
| 0 | 14 (50.0%) | 26 (72.2%) | |
| 1 | 10 (35.7%) | 8 (22.2%) | |
| 2 | 4 (14.3%) | 1 (2.8%) | |
| 3 | 0 (0.0%) | 1 (2.8%) | |
| Initial diagnosed site | | | <0.001 |
| GI tract | 20 (71.4%) | 0 (0.0%) | |
| lymph node | 5 (17.9%) | 24 (66.7%) | |
| others | 3 (10.7%) | 12 (33.3%) | |
| Presence of GI symptoms | | | |
| no | 18 (64.3%) | | |
| yes | 10 (35.7%) | | |
| Reason of endoscopic examinations | | | |
| abnormal findings at other imaging study | 14 (50.0%) | | |
| presence of GI symptoms | 8 (28.6%) | | |
| screening | 6 (21.4%) | | |
| Type of endoscopic examination | | | |
| esophagogastroduodenoscopy | 22 (78.6%) | 17 (47.2%) | 0.011 |
| colonoscopy | 26 (92.9%) | 12 (33.3%) | <0.001 |
| Endoscopic finding | | | |
| lymphomatous polyposis | 20 (71.4%) | | |
| polypoid mass | 6 (21.4%) | | |
| ulcerative lesion | 2 (7.1%) | | |
| Involved site of GI tract | | | |
| upper | 4 (14.3%) | | |
| lower | 15 (53.6%) | | |
| both | 9 (32.1%) | | |
| Ann Arbor stage | | | 0.012 |
| I | 0 (0.0%) | 1 (2.8%) | |
| II | 2 (7.1%) | 1 (2.8%) | |
| III | 0 (0.0%) | 11 (30.6%) | |
| IV | 26 (92.9%) | 23 (63.9%) | |
| IPI | | | 0.003 |
| low | 0 (0.0%) | 14 (38.9%) | |
| low-intermediate | 10 (35.7%) | 8 (22.2%) | |
| high-intermediate | 12 (42.9%) | 10 (27.8%) | |
| high | 6 (21.4%) | 4 (11.1%) | |
| MIPIb risk status | | | 0.040 |
| low | 5 (17.9%) | 9 (25.0%) | |
| intermediate | 5 (17.9%) | 15 (41.7%) | |
| high | 18 (64.3%) | 12 (33.3%) | |
| Positive Ki-67 status* | 21 (80.8%) | 22 (81.5%) | >0.999 |
| Elevated LDH, n (%) | 9 (32.1%) | 14 (38.9%) | 0.768 |
| BM involvement, n (%) | 18 (64.3%) | 16 (44.4%) | 0.185 |

*(Continued)*

**Table 1.** (Continued)

| N = 64 | GI-MCL | Non GI-MCL | p |
|---|---|---|---|
| | (n = 28) | (n = 36) | |
| Initial immunochemotherapy | | | 0.319 |
| R-CHOP | 22 (78.6%) | 23 (63.9%) | |
| R-HyperCVAD | 3 (10.7%) | 10 (27.8%) | |
| BR | 1 (3.6%) | 2 (5.6%) | |
| Other regimen[†] | 2 (7.1%) | 1 (2.8%) | |
| Auto PBSCT, n (%) | 6 (21.4%) | 10 (27.8%) | 0.771 |
| CR, n (%) | 27 (96.4%) | 32 (88.9%) | 0.519 |

[*]Assessed in Ki-67-evaluable patients (n = 27 in non GI-MCL group; n = 26 in GI-MCL group).

[†]Two of the GI-MCL patients were treated with R-FCM (rituximab-fludarabine, cyclophosphamide, and mitoxantrone) and R-MTX-ARAC (rituximab-methotrexate, cytarabine), respectively. One of the non GI-MCL patients were treated with R-FC (rituximab-fludarabine, cyclophosphamide).

GI, gastrointestinal; MCL, mantle cell lymphoma; ECOG, Eastern Cooperative Oncology Group; IPI, International Prognostic Index; MIPIb, Mantle Cell Lymphoma International Prognostic Index with biologic component; LDH, lactate dehydrogenase; BM, bone marrow; R-CHOP, rituximab-cyclophosphamide, doxorubicin, vincristine, prednisone; R-HyperCVAD, rituximab-hyperfractioned cyclophosphamide, vincristine, Adriamycin, dexamethasone; BR, bendamustine, rituximab; Auto PBSCT, autologous peripheral blood stem cell transplantation; CR, complete remission.

The rates of receiving autologous peripheral blood stem cell transplantation and achieving CR were also not different between the two groups.

## The role of endoscopy to detect GI involvement of MCL

Of the 28 patients in the GI-MCL group, 18 (64.3%) patients did not have GI symptoms. The most common endoscopic finding of MCL was lymphomatous polyposis (20/28, 71.4%). The lower GI tract (defined as the GI tract beyond the ligament of Treitz) was involved in 15 (53.6%) patients. In 9 (32.1%) patients, both the upper and lower GI tract were affected.

Table 2 summarizes the detection modalities of the 51 GI lesions in the GI-MCL group. CT scan could not detect 66.7% (34/51) of the lesions and PET-CT could not detect 33.3% (17/51) of lesions. Notably, 31.4% (16/51) of the lesions were detected only by endoscopic examination and were not detected by CT or PET-CT. Most were in the endoscopic finding of lymphomatous polyposis (87.5%, 14/16).

## Recurrence

Of the 27 patients achieving CR in GI-MCL group, 22 (81.5%) patients had recurred during a mean follow-up of 1.47 years. Of the 32 patients achieving CR in the non GI-MCL group, 18 (56.3%) patients recurred during a mean follow-up of 2.49 years. The cumulative incidence of recurrence was higher in the GI-MCL group compared with the non GI-MCL group but the difference was not statistically significant (P = 0.082) (Fig 2). Cumulative incidences of recurrence in the two groups at 1, 2, and 3 years were 33.9%, 60.2%, and 84.5% (95% CI, 13.1–49.8%, 34.8–75.7%, and 57.8–94.3%), respectively, in the GI-MCL group and 28.5%, 47.9%, and 51.9% (95% CI, 10.8–42.7%, 25.6–63.5%, and 29–67.4%), respectively, in the non GI-MCL group. In multivariate Cox regression analysis including the MCL group, age, sex, stage, IPI, MIPIb, immunochemotherapy, and autologous peripheral blood stem cell transplantation (auto PBSCT), stage (hazard ratio [HR] 1.994, 95% confidence interval [CI] 1.007–3.948) and auto PBSCT (HR 0.133, 95% CI 0.041–0.437) were identified as independent predictive factors for recurrence (Table 3).

**Table 2. Detection modalities and characteristics of 51 gastrointestinal lesions in 28 MCL patients with gastrointestinal tract involvement.**

| Detection modalities | | | N = 51 | Location | Endoscopic finding |
|---|---|---|---|---|---|
| Endoscopy | CT | PET-CT | | | |
| O* | O | O | 16 | Stomach: 2 | Lymphomatous polyposis: 7 |
| | | | | Duodenum: 1 | Polypoid mass: 7 |
| | | | | Small bowel: 0 | Ulcerative lesion: 2 |
| | | | | Ileocecal area: 7 | |
| | | | | Colon: 6 | |
| | | | | Rectum: 0 | |
| O | O | X† | 1 | Stomach: 0 | Lymphomatous polyposis: 1 |
| | | | | Duodenum: 0 | Polypoid mass: 0 |
| | | | | Small bowel: 0 | Ulcerative lesion: 0 |
| | | | | Ileocecal area: 0 | |
| | | | | Colon: 1 | |
| | | | | Rectum: 0 | |
| O | X | O | 18 | Stomach: 3 | Lymphomatous polyposis: 16 |
| | | | | Duodenum: 1 | Polypoid mass: 1 |
| | | | | Small bowel: 4 | Ulcerative lesion: 1 |
| | | | | Ileocecal area: 3 | |
| | | | | Colon: 3 | |
| | | | | Rectum: 4 | |
| O | X | X | 16 | Stomach: 4 | Lymphomatous polyposis: 14 |
| | | | | Duodenum: 1 | Polypoid mass: 2 |
| | | | | Small bowel: 0 | Ulcerative lesion: 0 |
| | | | | Ileocecal area: 4 | |
| | | | | Colon: 5 | |
| | | | | Rectum: 2 | |

* Type "O" means that the detection modality detected the relevant gastrointestinal lesions.

† Type "X" means that the detection modality could not detect the relevant gastrointestinal lesions.

MCL, mantle cell lymphoma; CT, computed tomography; PET-CT, Positron emission tomography-computed tomography.

### The role of endoscopy to detect MCL recurrence

Fig 3 summarizes the status of recurrence in the patients. All recurrences at the GI tract in both groups were macroscopic. Of the 22 patients with recurrence in the GI-MCL group, 13 (59.1%) patients recurred at the GI tract. Among them, recurrence was detected in 4 (4/13, 30.8%) patients only with endoscopic examinations. Radiologic examinations such as CT or PET-CT could not reveal recurrence in these patients.

Of the 18 patients with recurrence in the non GI-MCL group, 2 (11.1%) patients recurred de novo at the GI tract. Among them, recurrence was detected in 1 (1/2, 30.8%) patient only with endoscopic examinations.

### Survival

There were no significant differences in the overall survival between the two groups ($P = 0.793$) (Fig 4). Overall survival rates in the two groups at 1, 3, and 5 years were 96.4%, 87.2%, and 72.2% (95% CI, 89.8–100.0%, 74.6–100.0%, and 55.4–94.3%), respectively, in the GI-MCL group, and 97.2%, 81.2%, and 73.2% (95% CI, 92.0–100.0%, 68.6–96.1%, and 58.6–91.4%), respectively, in the non GI-MCL group.

# Recurrence

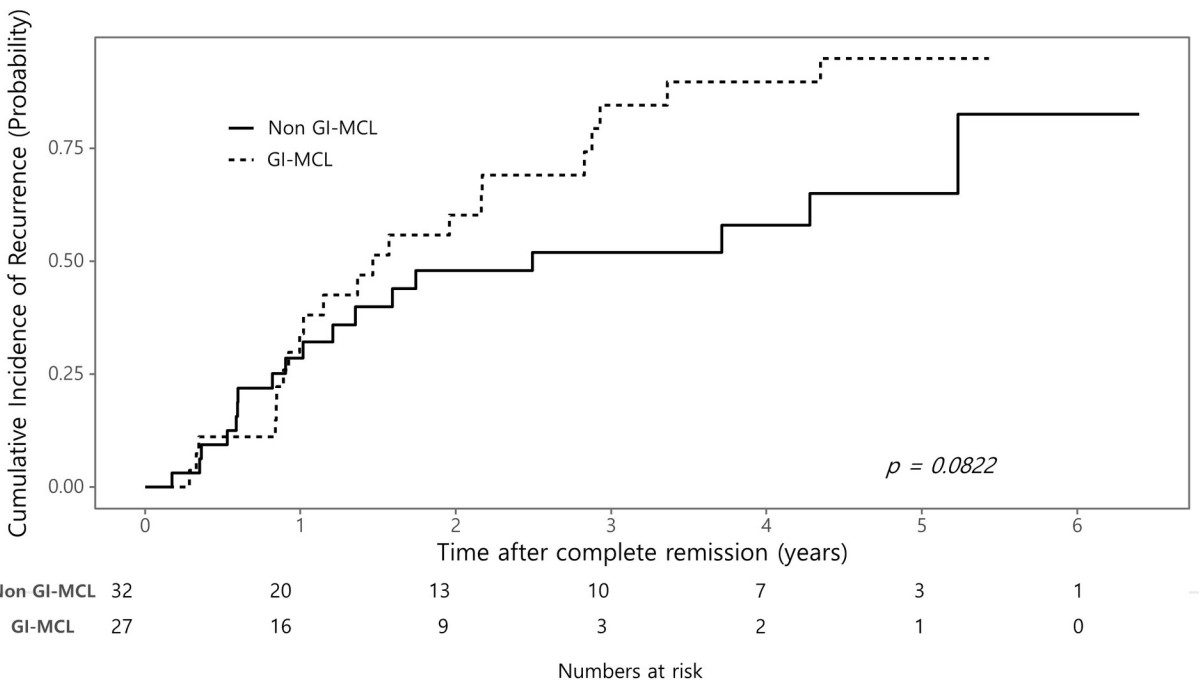

**Fig 2. Cumulative incidence of recurrence following complete remission in patients with mantle cell lymphoma according to gastrointestinal involvement.** Cumulative incidences of recurrence in the GI-MCL and non GI-MCL groups at 3 years were 94.8% and 64.9%, respectively.

## Discussion

Our results showed that MCL patients with GI tract involvement are older and usually have advanced stage and higher IPI score. Oncological outcomes including disease recurrence or

**Table 3. Sequential univariate and multivariate Cox proportional hazard regression models, showing independence of effect upon recurrence.**

| | Univariate | | | Multivariate (initial model) | | | Multivariate (final model) | | |
|---|---|---|---|---|---|---|---|---|---|
| | Hazard ratio | 95% CI | *P* | Hazard ratio | 95% CI | *P* | Hazard ratio | 95% CI | *P* |
| Group (ref. non GI-MCL group) | 1.740 | 0.920–3.274 | 0.086 | 1.328 | 0.505–3.494 | 0.565 | | | |
| Age (ref. ≤60 years) | 1.030 | 1.000–1.061 | 0.047 | 0.956 | 0.910–1.004 | 0.075 | 0.998 | 0.968–1.030 | 0.917 |
| Sex (ref. male) | 0.868 | 0.363–2.075 | 0.750 | | | | | | |
| Ann Arbor stage (ref. stage I) | 1.613 | 0.971–2.679 | 0.065 | 1.643 | 0.604–4.468 | 0.331 | 1.994 | 1.007–3.948 | 0.048 |
| IPI (ref. low risk) | 1.350 | 0.965–1.890 | 0.080 | 1.537 | 0.845–2.794 | 0.159 | 0.956 | 0.616–1.483 | 0.841 |
| MIPIb (ref. low risk) | 1.702 | 0.999–2.901 | 0.050 | 1.464 | 0.605–3.543 | 0.398 | | | |
| Immunochemotherapy (ref. R-CHOP) | 0.416 | 0.181–0.954 | 0.038 | 0.796 | 0.266–2.385 | 0.684 | | | |
| Auto PBSCT (ref. no) | 0.177 | 0.062–0.503 | 0.001 | 0.108 | 0.024–0.489 | 0.004 | 0.133 | 0.041–0.437 | <0.001 |

CI, confidence interval; GI, gastrointestinal; MCL, mantle cell lymphoma; IPI, International Prognostic Index; MIPIb, Mantle Cell Lymphoma International Prognostic Index with biologic component; R-CHOP, rituximab-cyclophosphamide, doxorubicin, vincristine, prednisone; Auto PBSCT, autologous peripheral blood stem cell transplantation.

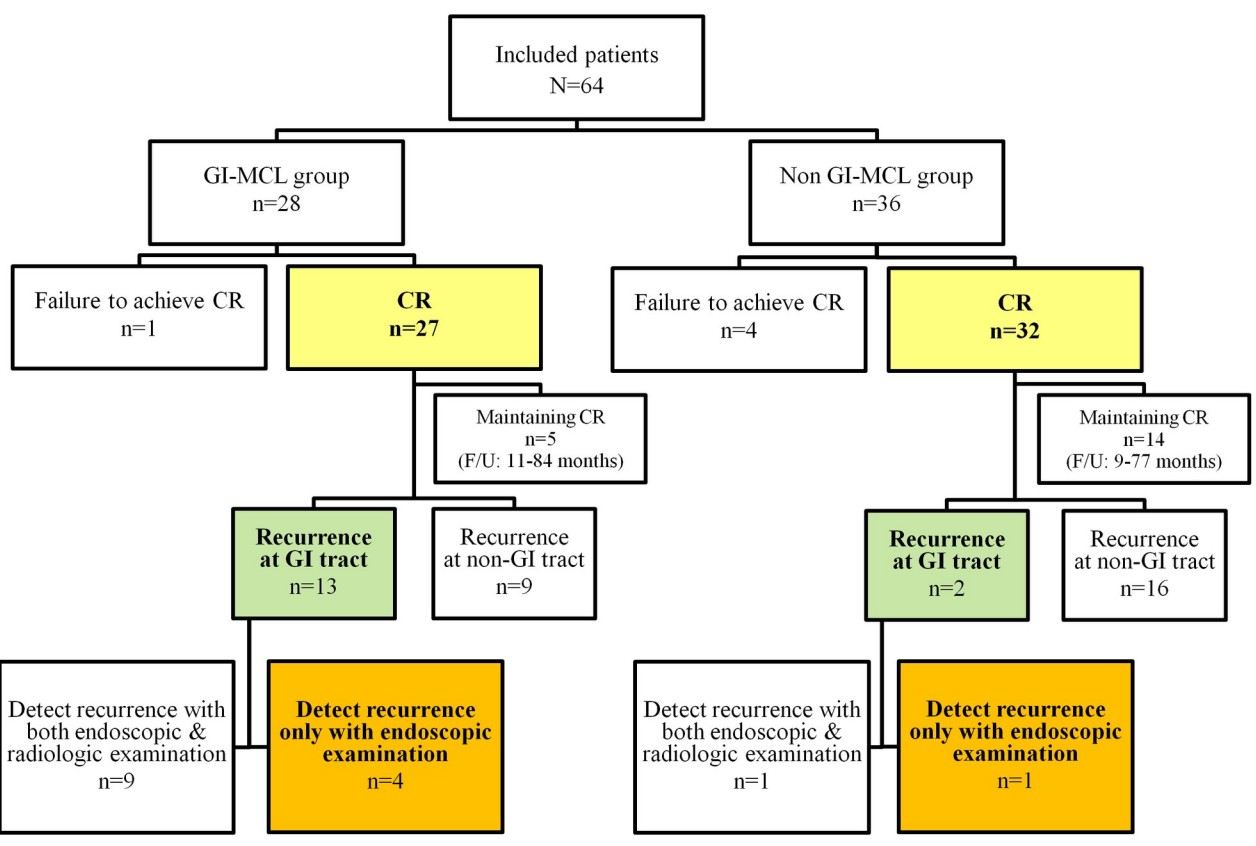

**Fig 3. Status of recurrence in the included patients.**

overall survival are not different according to GI tract involvement. We also found that initial endoscopic examinations at the time of diagnosis revealed a considerable number of lesions that did not cause symptoms and could not be detected by CT or PET-CT. In addition, endoscopic examinations detected subtle recurrences that CT or PET-CT did not identify during the follow-up of the MCL patients with GI tract involvement as well as those without GI tract involvement.

Macroscopic GI involvement has been reported in 15% to 30% of MCL patients at the time of diagnosis [13–18]. However, if endoscopic biopsies were performed on macroscopically normal mucosa as well as abnormal lesion, the percentage of cases with GI involvement could rise to about 90% [3, 8]. In our study, 43.8% (28/64) of newly diagnosed MCL patients had distinct GI tract involvement. The endoscopic examinations were performed mainly for abnormal findings in other imaging studies or for the purpose of screening because a large number of GI-MCL patients (64.3%, 18/28) did not have GI symptoms. This finding supports the NCCN clinical practice guideline that upper endoscopic or colonoscopic evaluation of the GI tract is necessary for confirmation of stage I–II disease [10]. Over 80% of the GI-MCL group showed lower GI tract involvement and the most common endoscopic finding was lymphomatous polyposis (71.4%, 20/28), which is consistent with the results of previous studies [2, 19–21].

The GI-MCL group had more advanced stage and higher IPI and MIPIb risk status compared with the non GI-MCL group, although the regimen of initial immunochemotherapy and auto PBSCT were not different between the two groups. The rates of achieving clinical

# Overall Survival

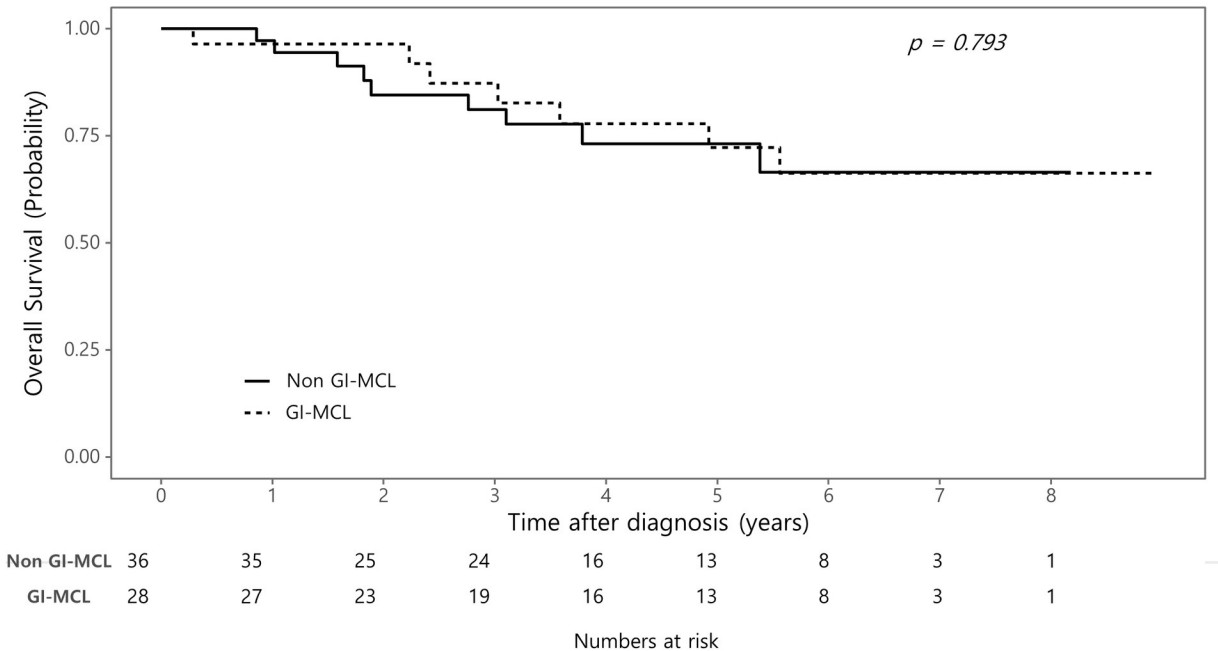

**Fig 4. Kaplan-Meier estimates of overall survival in patients with mantle cell lymphoma according to gastrointestinal involvement.**

remission were similar between the two groups in our study. Consistent with the aggressive nature of this type of lymphoma, about 70% of the patients achieving CR showed recurrence. Although the recurrence rate of the GI-MCL group was higher than that of the non GI-MCL group, the difference was not statistically significant. In addition, the presence of GI tract involvement was not the predictive factor for recurrence. These results correspond well with an earlier study stating that aggressive staging evaluation of the GI tract had little impact on patient management decisions [3]. In that study, the treatment was altered in only 3% of patients due to the GI tract involvement.

However, endoscopic examinations are indispensable for initial workup of MCL because they can expose the GI tract involvement that cannot be established by radiologic evaluation. In this study, approximately 70% of the GI-MCL group were pathologically diagnosed by endoscopic biopsy. We also found that more than 30% of GI lesions were detected only by endoscopic examinations and CT or PET-CT could not reveal them. While CT of chest, abdomen, and pelvis are routinely performed for staging procedure of MCL, the collapsed state of bowel loops makes diagnosis of GI lymphoma by CT difficult [11]. Evaluation of MCL by PET-CT is also hampered because the standard uptake values of involved sites often have low or intermediate values and it can be confused with bowel physiologic activity [22]. If clinicians miss gastrointestinal lesions because they do not perform endoscopic examination at the time of diagnosis, surveillance of recurrence depends only on radiological imaging, which makes early detection of recurrence difficult. Thus, we suggest that endoscopic examinations should

be performed during the initial workup of MCL, considering that mostly all MCL have microscopic GI tract involvement [3, 8].

The present study also revealed that upper endoscopy and colonoscopy should be performed to detect relapse for MCL patients who are followed up after clinical remission. More than half (13/22) of the GI-MCL group achieving CR relapsed at the GI tract. Interestingly, in four of the 13 patients, relapse was detected only with endoscopic examinations. CT scan and PET-CT could easily miss the subtle lesions on the GI tract. In the case of MCL without obvious GI tract involvement (the non GI-MCL group), endoscopic surveillances are also helpful to detect relapse. Our results showed that two of 18 patients with recurrence in the non GI-MCL group recurred de novo at the GI tract. In addition, relapse was detected in one patient only with endoscopic examinations. Therefore, if a recurrence is diagnosed first with an endoscopy before a radiological recurrence, salvage treatment for recurrence can be performed as quickly as possible.

Our opinion that endoscopic examinations should be performed during initial workup and surveillance of MCL patients is in disagreement with the earlier argument by Romaguera et al. [3]. The authors stated that surveillance colonoscopies were not warranted in MCL patients who have achieved CR because only 1 of 16 recurrences was detected by colonoscopic biopsy. However, a much larger number of cases of recurrence were confirmed by endoscopic examination alone in our study. If a clinician carries out regular inspections with the possibility of relapse in mind during follow up, endoscopic examinations may lead to early detection of MCL relapse and the relapse can be confirmed by tissue biopsy. In contrast, relying only on radiologic evaluation without regular endoscopic examination may overlook the progression of the recurred disease and can lead to an advanced relapse condition. In the case of MCL, the establishment of oncological assessment criteria based on endoscopy and preparation of guidelines for endoscopic surveillance are required and should be examined in a larger prospective study.

Clinical significance of GI involvement in MCL lymphoma seems to be underestimated. Relapsed MCL patients often show GI bleeding regardless of initial involvement of GI tract [23]. Active regular endoscopy can detect early relapse of GI lesions before GI bleeding occurs regardless of the presence of initial GI lesion. Bruton Tyrosine Kinase inhibitors such as ibrutinib seems to be highly effective for recurrent GI MCL as well as non GI involvement. However, since it has a hemorrhagic tendency, it can not be used immediately in patients with GI bleeding [24, 25].

There is something to be considered as a limitation of this retrospective cohort study. As shown in Table 1, the GI-MCL group received much more endoscopic examinations than the non GI-MCL group at the time of diagnosis and only less than half of the non GI-MCL group received endoscopic examinations. Therefore, the GI tract involvement of MCL might have been missed during the initial workup in the patients who did not undergo endoscopy examinations. This might result in underestimation of the GI involvement rate and may distort the analyses comparing between the GI-MCL and non GI-MCL groups. On the other hand, these results also reflect that endoscopic examinations were still omitted early in diagnosis in a considerable number of MCL patients and should be performed.

In conclusion, macroscopic GI tract involvement is present in nearly half of MCL patients at the time of diagnosis. These cases usually have advanced stage and higher IPI and MIPIb risk status. Routine upper endoscopic and colonoscopic examinations are recommended for initial staging regardless of GI symptoms. In addition, rigorous endoscopic surveillance for both upper and lower GI tract should be performed to detect subtle GI relapse during the follow-up of MCL patients with initial GI involvement who achieved remission. Even for the

patients with no initial GI involvement, it is advisable to conduct surveillance endoscopy regularly as there is a possibility of a new recurrence on the GI tract.

## Supporting information

**S1 Table. Data set of the included mantle cell lymphoma patients.**
(CSV)

## Author Contributions

**Conceptualization:** Han Hee Lee, Seok-Goo Cho, In Seok Lee, Seung Eun Jung, Kyung-Sin Park.

**Data curation:** Han Hee Lee, Hye Jin Cho.

**Formal analysis:** Han Hee Lee, Hye Jin Cho, Young-Woo Jeon.

**Funding acquisition:** Han Hee Lee.

**Investigation:** Hye Jin Cho, Young-Woo Jeon, Joo Hyun O, Seung Eun Jung, Byung Ock Choi, Kyung-Sin Park, Suk-Woo Yang.

**Methodology:** Han Hee Lee.

**Supervision:** Seok-Goo Cho, In Seok Lee.

**Writing – original draft:** Han Hee Lee.

**Writing – review & editing:** Han Hee Lee, Seok-Goo Cho, In Seok Lee, Young-Woo Jeon, Joo Hyun O, Seung Eun Jung, Byung Ock Choi, Kyung-Sin Park, Suk-Woo Yang.

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
