## [Decision Letter · Decision Letter 0]

13 Jul 2020

PONE-D-20-04372

Mantle Cell Lymphoma with Gastrointestinal Involvement and the Role of Endoscopic Examinations

PLOS ONE

Dear Dr. Cho,

Thank you for submitting your manuscript to PLOS ONE. After careful consideration, we feel that it has merit but does not fully meet PLOS ONE’s publication criteria as it currently stands. Therefore, we invite you to submit a revised version of the manuscript that addresses the points raised during the review process. In particular, as noted in the comments by reviewer 2, authors need to clarify whether the patients had standard or other treatment regimens for MCL and if so, these data should be included in the analyses.

If you would like to make changes to your financial disclosure,
please include your updated statement in your cover letter. Guidelines for resubmitting your figure
files are available below the reviewer comments at the end of this letter.

We look forward to receiving your revised manuscript.

Kind regards,

Arun Rishi, Ph.D.

Academic Editor

PLOS ONE

Journal Requirements:

2.  In the ethics statement in the manuscript and in the online submission form, please provide additional information about the patient records used in your retrospective study, including: a) whether all data were fully anonymized before you accessed them and b) the date range (month and year) during which patients' medical records were accessed."

3. Thank you for including your ethics statement:

'The Institutional Review Board of our institution approved this study (KC11RISI0983)'.

a.Please amend your current ethics statement to include the full name of the ethics committee/institutional review board(s) that approved your specific study.  

b.Once you have amended this statement in the Methods section of the manuscript, please add the same text to the “Ethics Statement” field of the submission form (via “Edit Submission”).

For additional information about PLOS ONE ethical requirements for human subjects research, please refer to " ext-link-type="uri" xlink:type="simple">http://journals.plos.org/plosone/s/submission-guidelines#loc-human-subjects-research."

4. Please provide a sample size and power calculation in the Methods, or discuss the reasons for not performing one before study initiation."

5. In the Discussion section, please include a section discussing any potential limitations in your retrospective study.

Reviewers' comments:

Reviewer's Responses to Questions

**Comments to the Author**

1. Is the manuscript technically sound, and do the data support the conclusions?

Reviewer #1: Yes

Reviewer #2: Yes

2. Has the statistical analysis been performed appropriately and rigorously? 

Reviewer #1: Yes

Reviewer #2: No

3. Have the authors made all data underlying the findings in their manuscript fully available?

Reviewer #1: Yes

Reviewer #2: Yes

4. Is the manuscript presented in an intelligible fashion and written in standard English?

Reviewer #1: Yes

Reviewer #2: Yes

5. Review Comments to the Author

Reviewer #1: PONE-D-20-04372

Research Article

Mantle Cell Lymphoma with Gastrointestinal Involvement and the Role of Endoscopic Examinations

This is a well-written paper I have no hesitation in recommending it for publication following some minor tidy-up. The focus of the paper is well phrased.

The authors state, "We also found that initial endoscopic examinations at the time of diagnosis revealed a considerable number of lesions that did not cause symptoms and could not be detected by CT or PET-CT." Therefore, gastrointestinal involvement with mantle cell lymphoma might have been missed during the initial workup in patients who did not undergo endoscopy examinations. This results in underestimation of GI involvement rate and may distort the analyses comparing between the GI-MCL and non GI-MCL groups.

1-1) It is recommended that the numbers of patients who underwent esophagogastroduodenoscopy, colonoscopy, and small intestinal endoscopy in both GI-MCL and non GI-MCL groups be specified in the Table 1.

1-2) This issue should be mentioned as one of the limitations of this study.

Reviewer #2: This paper by Han Hee Lee et al describes a series of patients with mantle cell lymphoma (MCL) and the value of studying the involvement of the gastrointestinal (GI) tract. This is a retrospective study in a single center that includes 74 patients with MCL of whom 64 are analyzed. The description of the macroscopic involvement of the GI is clearly described, being the most interesting point of the study. Of value, they described the characteristics of the GI lesions (N=51) and how they are recognized by 3 different techniques (endoscopy, TC and PET). Finally, outcome impact of the GI involvement is analyzed in a univariate and multivariate analysis and here I see some weaknesses.

However, some points need to be addressed:

Major issues:

Staging studies are very carefully described at recurrence, but not at diagnosis. Please, add and describe these procedures at diagnosis, in particular imaging techniques.

Type of induction immunochemotherapy should be briefly described, for both young and elderly patients. In addition, the number of patients who are transplanted is very low (only 16 out of 64), and this is surprising due to the median age of the study population.

Reason for endoscopic examinations were: abnormal findings at other imaging study, presence of GI symptoms, screening. Do all patients without GI symptoms or normal CTs go for GI endoscopies?

How often was GI examination during follow-up? Does it include upper and lower endoscopy in all cases?. I am surprised by the high number of recurrences in the GI group, and even with a short follow-up. This is unexpected in my experience, especially in the rituximab era.

According to the results of the paper, I will recommend upper endoscopy and colonoscopy to detect relapse for MCL patients but preferentially in those patients who had GI at diagnosis (22 cases: 81.5%). The low GI relapse rate in no-GI patients (2 cases: 11.1%) reduces the utility of GI follow-up in this grup. The authors recommend GI for all patients in follow-up. This should be revised in the paper.

I understand that the GI recurrences were all macroscopic (and not microscopic). Is this right? Please, clarify.

I miss type of treatment and maintenance with rituximab in the univariate (and multivariate) analysis for recurrence. I understand that this is a very important point because type treatment is totally relevant for outcome in MCL. In other words, outcome clearly depends on regimen type: R-CVP is worse than R-CHOP/R-DHAP. 19 patients were treated with “other regimens”. It would be interesting to know the regimen. Moreover, I would like to know if R-CHOP also includes patients who were treated with high doses of cytarabine. I believe very relevant the evaluation of the type of treatment in outcome (recurrence and survival) and this is more relevant than the variable transplantation, and this variable is included. Please, review.

Conclusions are a bit strong considering the retrospective and relatively small number of cases. Please, review them.

Minor issues:

Table 2: O might be misleading for the reader: please, clarify.

Pg. 13: medians and interquartile for age, but in table 1 you use median and range.

Survival time is expressed in days, but in lymphoma is usually calculated in months or years.

The value of these findings could be different in the era of BTK-inhibitors. A comment might be of interest in the discussion.

I also miss a paragraph of limitations of the study.

6. PLOS authors have the option to publish the peer review history of their article (what does this mean?). If published, this will include your full peer review and any attached files.

Reviewer #1: No

Reviewer #2: **Yes: **Antonio Salar

---

## [Author Response · Author response to Decision Letter 0]

26 Aug 2020

Reviewers' comments:

Review Comments to the Author

Reviewer #1: PONE-D-20-04372

This is a well-written paper I have no hesitation in recommending it for publication following some minor tidy-up. The focus of the paper is well phrased.

The authors state, "We also found that initial endoscopic examinations at the time of diagnosis revealed a considerable number of lesions that did not cause symptoms and could not be detected by CT or PET-CT." Therefore, gastrointestinal involvement with mantle cell lymphoma might have been missed during the initial workup in patients who did not undergo endoscopy examinations. This results in underestimation of GI involvement rate and may distort the analyses comparing between the GI-MCL and non GI-MCL groups.

1-1) It is recommended that the numbers of patients who underwent esophagogastroduodenoscopy, colonoscopy, and small intestinal endoscopy in both GI-MCL and non GI-MCL groups be specified in the Table 1.

1-2) This issue should be mentioned as one of the limitations of this study.

Thank you so much for your kind opinion and understanding the significance of our research.

For the comment 1-1, following your advice, we specified the numbers of patients who underwent esophagogastroduodenoscopy and colonoscopy in both groups in the Table (page 19). We could easily confirm that the GI-MCL group received much more endoscopic examinations than the non GI-MCL group at the time of diagnosis and only less than half of the non GI-MCL group received endoscopic examinations. Small intestine endoscopy such as capsule endoscopy or enteroscopy was not performed for initial staging. 

For the comment 1-2, we mentioned the above limitation in the “Discussion” section as follows (page 13):

There is something to be considered as a limitation of this retrospective cohort study. As shown in Table 1, the GI-MCL group received much more endoscopic examinations than the non GI-MCL group at the time of diagnosis and only less than half of the non GI-MCL group received endoscopic examinations. Therefore, the GI tract involvement of MCL might have been missed during the initial workup in the patients who did not undergo endoscopy examinations. This might result in underestimation of the GI involvement rate and may distort the analyses comparing between the GI-MCL and non GI-MCL groups. On the other hand, these results also reflect that endoscopic examinations were still omitted early in diagnosis in a considerable number of MCL patients and should be performed.

Reviewer #2: This paper by Han Hee Lee et al describes a series of patients with mantle cell lymphoma (MCL) and the value of studying the involvement of the gastrointestinal (GI) tract. This is a retrospective study in a single center that includes 74 patients with MCL of whom 64 are analyzed. The description of the macroscopic involvement of the GI is clearly described, being the most interesting point of the study. Of value, they described the characteristics of the GI lesions (N=51) and how they are recognized by 3 different techniques (endoscopy, TC and PET). Finally, outcome impact of the GI involvement is analyzed in a univariate and multivariate analysis and here I see some weaknesses.

However, some points need to be addressed:

Major issues:

Staging studies are very carefully described at recurrence, but not at diagnosis. Please, add and describe these procedures at diagnosis, in particular imaging techniques.

Thank you for your kind advice. 

When the patients were confirmed as MCL pathologically, our Catholic University Lymphoma Group of Seoul St. Mary’s Hospital follows the current standard staging workup of MCL. Following your advice, we added the related sentence in the “Method - Patients and Eligibility Criteria” section as follows (page 5):

When the patients were confirmed as MCL pathologically, they underwent the current standard staging workup that included a physical examination, blood cell counts, routine blood chemistries, computed tomography (CT) of the chest, abdomen, and pelvis, [18F]fluorodeoxyglucose-positron emission tomography CT (FDG-PET CT) and a bone marrow evaluation.

Type of induction immunochemotherapy should be briefly described, for both young and elderly patients. In addition, the number of patients who are transplanted is very low (only 16 out of 64), and this is surprising due to the median age of the study population.

Thank you for your comment. 

We also follows the current standards of care of MCL. Following your advice, we added the related sentence in the “Method - Treatment and follow-up” section as follows (page 6):

The first line regimens for elderly patients included rituximab plus cyclophosphamide/doxorubicin/oncovin/prednisone (R-CHOP) and bendamustine/ rituximab (BR). For younger patients, rituximab plus hyperfractioned cyclophosphamide/vincristine/adriamycin/dexamethasone (R-HyperCVAD) or other modified regimens were considered.

As you have pointed out, we performed autologous peripheral blood stem cell transplantation (auto-PBSCT) in a small number of patients due to the limitation of choosing the first-line regimen for MCL patients. In our country we commonly used R-CHOP as the first-line regimen. However as you know it is hard to achieve the deep response before auto-PBSCT. Nordic regimen such as R-DHAP or alternative regimen are not covered by public medical insurance in Korea. Among high-dose ARA-C containing rituximab-based regimen, R-HyperCVAD is only covered by medical insurance in Korea, but it seems to be more toxic compared to R-DHAP, so it is not commonly used.

Reason for endoscopic examinations were: abnormal findings at other imaging study, presence of GI symptoms, screening. Do all patients without GI symptoms or normal CTs go for GI endoscopies?

Thank you for your question. 

Not all patients received a GI endoscopy just because they had been diagnosed with MCL. Some did and some not. Therefore, GI involvement with MCL might have been missed during the initial workup in patients who did not undergo endoscopic examinations. If all patients went for GI endoscopies, we could have revealed a considerable number of lesions that did not cause symptoms and could not be detected by CT or PET-CT. That’s the point that we want to emphasize in this study. 

To clarify this, we specified the numbers of patients who underwent esophagogastroduodenoscopy and colonoscopy in both groups in the Table (page 19). Also we mentioned this limitation in the “Discussion” section as follows (Page 13):

There is something to be considered as a limitation of this retrospective cohort study. As shown in Table 1, the GI-MCL group received much more endoscopic examinations than the non GI-MCL group at the time of diagnosis and only less than half of the non GI-MCL group received endoscopic examinations. Therefore, the GI tract involvement of MCL might have been missed during the initial workup in the patients who did not undergo endoscopy examinations. This might result in underestimation of the GI involvement rate and may distort the analyses comparing between the GI-MCL and non GI-MCL groups. On the other hand, these results also reflect that endoscopic examinations were still omitted early in diagnosis in a considerable number of MCL patients and should be performed.

How often was GI examination during follow-up? Does it include upper and lower endoscopy in all cases?. I am surprised by the high number of recurrences in the GI group, and even with a short follow-up. This is unexpected in my experience, especially in the rituximab era.

Thank you for your question. 

There are no standard guidelines recommending regular GI endoscopic examinations for MCL patients, but we conduct both upper and lower endoscopy every three to six months for the first year of GI-MCL patients and every six to twelve months thereafter by increasing the gap. Even for non-GI MCL patients, we are trying to conduct surveillance endoscopy as often as possible. In addition, although R-CHOP generally improve clinical outcome of MCL, I think it seems to be difficult to induce long-term remission in gastrointestinal MCL lymphoma. I think gastrointestinal involvement will be an important hidden poor prognostic factor in MCL although there are few reports related to gastrointestinal MCL. These findings will be elucidated through prospective research.

According to the results of the paper, I will recommend upper endoscopy and colonoscopy to detect relapse for MCL patients but preferentially in those patients who had GI at diagnosis (22 cases: 81.5%). The low GI relapse rate in no-GI patients (2 cases: 11.1%) reduces the utility of GI follow-up in this grup. The authors recommend GI for all patients in follow-up. This should be revised in the paper.

Thank you for your kind comment. 

Following your advice, we modified the related sentence in the “Discussion” section as follows (page 13):

In addition, endoscopic surveillance should be performed for all MCL patients achieving remission regardless of GI tract involvement. rigorous endoscopic surveillance for both upper and lower GI tract should be performed to detect subtle GI relapse during the follow-up of MCL patients with initial GI involvement who achieved remission. Even for the patients with no initial GI involvement, it is advisable to conduct surveillance endoscopy regularly as there is a possibility of a new recurrence on the GI tract.

I understand that the GI recurrences were all macroscopic (and not microscopic). Is this right? Please, clarify.

Yes, you are right. All GI recurrences were macroscopic. To clarify this, we added the related sentence in the “Result - The role of endoscopy to detect MCL recurrence” section as follows (page 9):

Figure 3 summarizes the status of recurrence in the patients. All recurrences at the GI tract in both groups were macroscopic. Of the 22 patients with recurrence in the GI-MCL group, 13 (59.1%) patients …

I miss type of treatment and maintenance with rituximab in the univariate (and multivariate) analysis for recurrence. I understand that this is a very important point because type treatment is totally relevant for outcome in MCL. In other words, outcome clearly depends on regimen type: R-CVP is worse than R-CHOP/R-DHAP. 19 patients were treated with “other regimens”. It would be interesting to know the regimen. Moreover, I would like to know if R-CHOP also includes patients who were treated with high doses of cytarabine. I believe very relevant the evaluation of the type of treatment in outcome (recurrence and survival) and this is more relevant than the variable transplantation, and this variable is included. Please, review.

Thank you for your comment. 

Following your comment, we described the treatment regimens more concretely in Table 1. After R-CHOP, R-hyperCVAD was the most frequently treated, followed by BR therapy.

These regimens were initial immunochemotherapy for chemo-naïve MCL patients. So, R-CHOP did not include the patients who were treated with high doses of cytarabine before.

Also, we re-analyzed the multivariate Cox proportional hazard regression models to identify the predictive factors for recurrence after including the type of immunochemotherapy (R-CHOP vs. the others). As the result, the type of immunochemotherapy was not the independent predictive factor for recurrence (HR 0.796, 95% CI 0.266–2.385). To clarify this, we modified the Table 3. 

Conclusions are a bit strong considering the retrospective and relatively small number of cases. Please, review them.

Thank you for your considerate comment. Following your advice, we modified the conclusion paragraph in the “Discussion” section as follows (page 13):

In conclusion, macroscopic GI tract involvement is present in nearly half of MCL patients at the time of diagnosis. These cases usually have advanced stage and higher IPI and MIPIb risk status. Routine upper endoscopic and colonoscopic examinations are recommended for initial staging regardless of GI symptoms. In addition, endoscopic surveillance should be performed for all MCL patients achieving remission regardless of GI tract involvement. rigorous endoscopic surveillance for both upper and lower GI tract should be performed to detect subtle GI relapse during the follow-up of MCL patients with initial GI involvement who achieved remission. Even for the patients with no initial GI involvement, it is advisable to conduct surveillance endoscopy regularly as there is a possibility of a new recurrence on the GI tract.

Minor issues:

Table 2: O might be misleading for the reader: please, clarify.

Thank you for your kind advice.

Type “O” or “X” means whether the detection modality detected the relevant GI lesions. To clarify this, we made footnotes about this under the Table 2. 

Pg. 13: medians and interquartile for age, but in table 1 you use median and range.

Thank you for your kind correction. We simply corrected the related sentence in the manuscript. 

Survival time is expressed in days, but in lymphoma is usually calculated in months or years.

Thank you for your advice. Following your advice, we modified the measure of survival time from days to years in the relevant figures. 

The value of these findings could be different in the era of BTK-inhibitors. A comment might be of interest in the discussion.

Thank you for your comment.

As you mentioned ibrutinib is a very important approach for relapsed MCL patients. So we revised the Discussion section as follows (page 13). 

Clinical significance of GI involvement in MCL lymphoma seems to be underestimated. Relapsed MCL patients often show GI bleeding regardless of initial involvement of GI tract. Active regular endoscopy can detect early relapse of GI lesions before GI bleeding occurs regardless of the presence of initial GI lesion. Bruton Tyrosine Kinase inhibitors such as ibrutinib seems to be highly effective for recurrent GI MCL as well as non GI involvement. However, since it has a hemorrhagic tendency, it can not be used immediately in patients with GI bleeding.

I also miss a paragraph of limitations of the study.

Thank you for your comment. We added the paragraph regarding the limitation of our study in the “Discussion” section as follows (page 13):

There is something to be considered as a limitation of this retrospective cohort study. As shown in Table 1, the GI-MCL group received much more endoscopic examinations than the non GI-MCL group at the time of diagnosis and only less than half of the non GI-MCL group received endoscopic examinations. Therefore, the GI tract involvement of MCL might have been missed during the initial workup in the patients who did not undergo endoscopy examinations. This might result in underestimation of the GI involvement rate and may distort the analyses comparing between the GI-MCL and non GI-MCL groups. On the other hand, these results also reflect that endoscopic examinations were still omitted early in diagnosis in a considerable number of MCL patients and should be performed.

---

## [Decision Letter · Decision Letter 1]

14 Sep 2020

Mantle Cell Lymphoma with Gastrointestinal Involvement and the Role of Endoscopic Examinations

PONE-D-20-04372R1

Dear Dr. Cho,

We’re pleased to inform you that your manuscript has been judged scientifically suitable for publication and will be formally accepted for publication once it meets all outstanding technical requirements.

Kind regards,

Arun Rishi, Ph.D.

Academic Editor

PLOS ONE

Additional Editor Comments (optional):

Reviewers' comments:

Reviewer's Responses to Questions

**Comments to the Author**

1. If the authors have adequately addressed your comments raised in a previous round of review and you feel that this manuscript is now acceptable for publication, you may indicate that here to bypass the “Comments to the Author” section, enter your conflict of interest statement in the “Confidential to Editor” section, and submit your "Accept" recommendation.

Reviewer #1: All comments have been addressed

2. Is the manuscript technically sound, and do the data support the conclusions?

Reviewer #1: Yes

3. Has the statistical analysis been performed appropriately and rigorously? 

Reviewer #1: Yes

4. Have the authors made all data underlying the findings in their manuscript fully available?

Reviewer #1: Yes

5. Is the manuscript presented in an intelligible fashion and written in standard English?

Reviewer #1: Yes

6. Review Comments to the Author

Reviewer #1: "Mantle Cell Lymphoma with Gastrointestinal Involvement and the Role of Endoscopic Examinations" (PONE-D-20-04372R1).

The authors have revised the manuscript according to the comments of the reviewers.

7. PLOS authors have the option to publish the peer review history of their article (what does this mean?). If published, this will include your full peer review and any attached files.

Reviewer #1: No

---

## [Editor Report · Acceptance letter]

17 Sep 2020

PONE-D-20-04372R1 

Mantle Cell Lymphoma with Gastrointestinal Involvement and the Role of Endoscopic Examinations 

Dear Dr. Cho:

I'm pleased to inform you that your manuscript has been deemed suitable for publication in PLOS ONE. Congratulations! Your manuscript is now with our production department. 

Kind regards, 

on behalf of

Prof Arun Rishi 

Academic Editor

PLOS ONE